# Ligand discrimination in immune cells: Signal processing insights into immune dysfunction in ER+ breast cancer

Adina Matache[1‡], Joao Rodrigues Lima-Junior[2‡], Maxim Kuznetsov[1], Konstancja Urbaniak[1], Sergio Branciamore[1], Andrei S. Rodin[1], Peter P. Lee[2*], Russell C. Rockne [1*]

1 Department of Computational and Quantitative Medicine, Beckman Research Institute of City of Hope, Duarte, California, United States of America, 2 Department of Immuno-Oncology, Beckman Research Institute of City of Hope, Duarte, California, United States of America

‡ co-first authors
* rrockne@coh.org (RCR), plee@coh.org (PPL)

## Abstract

Prior studies have shown that approximately 40% of estrogen receptor positive (ER+) breast cancer (BC) patients harbor immune signaling defects in their blood at diagnosis, and the presence of these defects predicts overall survival. Therefore, it is of interest to quantitatively characterize and measure signaling errors in immune signaling systems in these patients. Here we propose a novel approach combining communication theory and signal processing concepts to model ligand discrimination in immune cells in the peripheral blood. We use the model to measure the specificity of ligand discrimination in the presence of molecular noise by estimating the probability of error, which is the probability of making a wrong ligand identification. We apply our model to the JAK/STAT signaling pathway using high dimensional spectral flow cytometry measurements of transcription factors, including phosphorylated STATs and SMADs, in immune cells stimulated with several cytokines (IFNγ, IL-2, IL-6, IL-4, and IL-10) from 19 ER+ breast cancer patients and 32 healthy controls. In addition, we apply our model to 10 healthy donor samples treated with a clinically approved JAK1/2 inhibitor. Our results show reduced ligand identification accuracy and higher levels of molecular noise in BC patients as compared to healthy controls, which may indicate altered immune signaling and the potential for immune cell dysfunction in these patients. Moreover, the inhibition of JAK1/2 produces a unique pattern of signaling dysfunction, inducing increased ligand detection error rates and reduced signal-to-noise ratios for most immune cell subtypes. These results suggest a means to improve the use of signaling kinase inhibitor therapies by identifying patients with favorable ligand discrimination specificity profiles in their immune cells.

**Data availability statement:** All relevant data and computational codes to reproduce all results are provided at https://github.com/rrockne/BreastCancerSignalProcessing, https://doi.org/10.5281/zenodo.16906422.

**Funding:** Research reported in this publication included work performed in the Analytical Cytometry and Biostatistics and Mathematical Oncology Shared Resources supported by the National Cancer Institute of the National Institutes of Health under grant numbers P30CA033572, U01CA232216. The content is solely the responsibility of the authors and does not necessarily represent the official views of the National Institutes of Health. The funders had no role in study design, data collection and analysis, decision to publish, or preparation of the manuscript. All authors received salary support from NIH grant number U01CA232216.

**Competing interests:** The authors have declared that no competing interests exist.

## Author summary

Approximately 40% of estrogen receptor-positive breast cancer patients have problems in immune cell signaling at diagnosis, which can affect survival. This study introduces a new method using ideas from communication theory to understand how immune cells recognize signals (called ligands) in the blood. By modeling how accurately cells identify these signals despite the presence of molecular "noise," we were able to measure the likelihood of errors in signal detection in immune cells. We applied this communication model to analyze blood samples collected from breast cancer patients and healthy individuals, to reveal how cells respond to various immune-stimulating molecules. The results showed that breast cancer patients had more difficulty correctly identifying signals, suggesting their immune systems may not function properly. Additionally, when healthy cells were treated with a drug that blocks certain signaling pathways (JAK1/2), they showed similar issues in correctly identifying signals. These findings could help tailor treatments by identifying patients whose immune cells are better at signal recognition and could lead to new perspectives on the causes and consequences of immune dysfunction in breast cancer.

## Introduction

Precise and efficient communication between immune cells is essential for maintaining homeostasis and for mounting effective immune responses to pathogens. Cytokine-mediated signaling is a fundamental mechanism through which cells exchange information, that can be conceptualized as a communication network where cells act as both senders and receivers of signals. Here we apply concepts of information transmission, reception, and processing from digital communication theory to quantitatively study the function—and dysfunction—of immune signaling networks in healthy homeostasis and in breast cancer. Previous studies have shown that approximately 40% of estrogen receptor positive (ER+) breast cancer (BC) patients harbor immune signaling defects in their blood at diagnosis, and the presence of these defects predicts overall survival [1–3]. It is therefore of interest to quantitatively characterize and measure signaling errors and molecular noise in immune signaling systems.

Cell signaling is composed of several precisely regulated steps: cells release signaling molecules, such as cytokines, which diffuse between cells to interact with receptors on target cells. Receptor engagement initiates a cascade of intracellular events, including phosphorylation of proteins, that act in concert to determine cellular function or behavior. These processes mirror key principles of communication theory, including signal generation, message encoding, transmission, and reception, which are essential components for understanding how information is conveyed and interpreted between cells, particularly within the immune system. Cells encode signals from ligands through STAT phosphorylation. For example, IL-10 stimulation results in

STAT3 phosphorylation, IL-2 induces STAT5 phosphorylation, and STAT3 is strongly phosphorylated by IL-6 [4–7]. However, we find that each cytokine induces a pattern of phosphorylation across multiple STATs/SMADs and these patterns encode information that enable ligand discrimination [8].

Integrating digital communication theory into the study of cell signaling provides a mathematically rigorous and structured approach to studying cell signal processing. By examining the probability of signaling error and signal-to-noise ratio, we can gain insights into the efficiency and accuracy of cellular communication in immune cells from healthy donors versus patients with breast cancer. This interdisciplinary perspective not only enhances our understanding of signal transduction and cellular response mechanisms but also has significant implications for advancing research in immunology, and cell biology, and for therapeutic interventions which may interfere with, or may be designed to correct, errors in cell signaling. Here we study how communication theory principles can enhance our understanding of cell signaling alterations in peripheral blood immune cells from ER+ breast cancer, as compared to healthy donors.

## Communication model

A growing number of studies have used information-theoretic approaches to analyze complex single-cell data and to understand reliable communication in cell populations [9–14], and to compile a compendium of responses to cytokine stimuli [15]. Shannon's information theory [16], originally developed for digital communication systems, established fundamental limits on communication and information transmission. In Shannon's formulation, the basic problem of reliable transmission of information is stated in statistical terms, using probabilistic models for information sources and communication channels. At a high level, all communication systems have an information source (input) which sends a message, encoded or mapped into a unique signal by a transmitter. The signal is then transported from the transmitter to the receiver through a channel. Regardless of the physical medium used for transmission of the information, the main feature of the channel is that the transmitted signal is corrupted in a random manner by a variety of mechanisms, such as additive noise. The receiver decodes the corrupted signal into a decoded message, which is consumed in turn by an information sink (output). Under the principles of information theory, reliable transmission of information is possible if the information rate from the source is less than the channel capacity, which is the maximal mutual information (MI) between the input and the output of the channel.

This communication model is general enough to be applied to systems other than digital communication systems, such as a cell communication system (Fig 1). An intuitive application to cellular signaling is that the source is a cell that secretes a cell-signaling protein or cytokine (the message), recognized by a receptor (the transmitter). The receptor initiates a signaling pathway (the channel), which results in the activation of a transcription factor (the receiver). The transcription factor in turn induces gene expression (the output). When cells are stimulated with different cytokines, each cell may respond differently due to molecular noise, variations in ligand-receptor affinity, and other factors.

Here we focus on the JAK/STAT intracellular signaling axis [5], with each phosphorylation event of a STAT (pSTAT) or SMAD (pSMAD) molecule modeled as a random variable with an empirical distribution. Under a Gaussian distribution assumption, the problem of signal identification is similar to the problem of signal detection in digital communications, where the noise is assumed to be an additive, zero-mean Gaussian noise [17]. The main difference is that in digital communication systems, the set of possible transmitted signals is known, while in cellular communication systems, the signals carried by cytokines are unknown. Therefore, we must first define the cytokine-induced signals from the responses determined by the phosphorylation status of the STATs. With a zero-mean Gaussian noise assumption, the signal is given by the mean of the pSTAT responses; therefore, the Gaussian assumption is conveniently apropos. Then, the basic communication model for immune signaling in the JAK/STAT signaling pathway consists of signal generation, noisy channel transmission, and signal detection. The signal generation model maps each cytokine-induced signal to a multi-dimensional vector with elements equal to the means of the transcription factor responses. This vector defines a unique pattern of response for each cytokine treatment, which we call a *codeword*. The noisy channel adds a zero-mean Gaussian noise to the signaled codeword, and the detector performs optimal signal detection and identification, assuming the

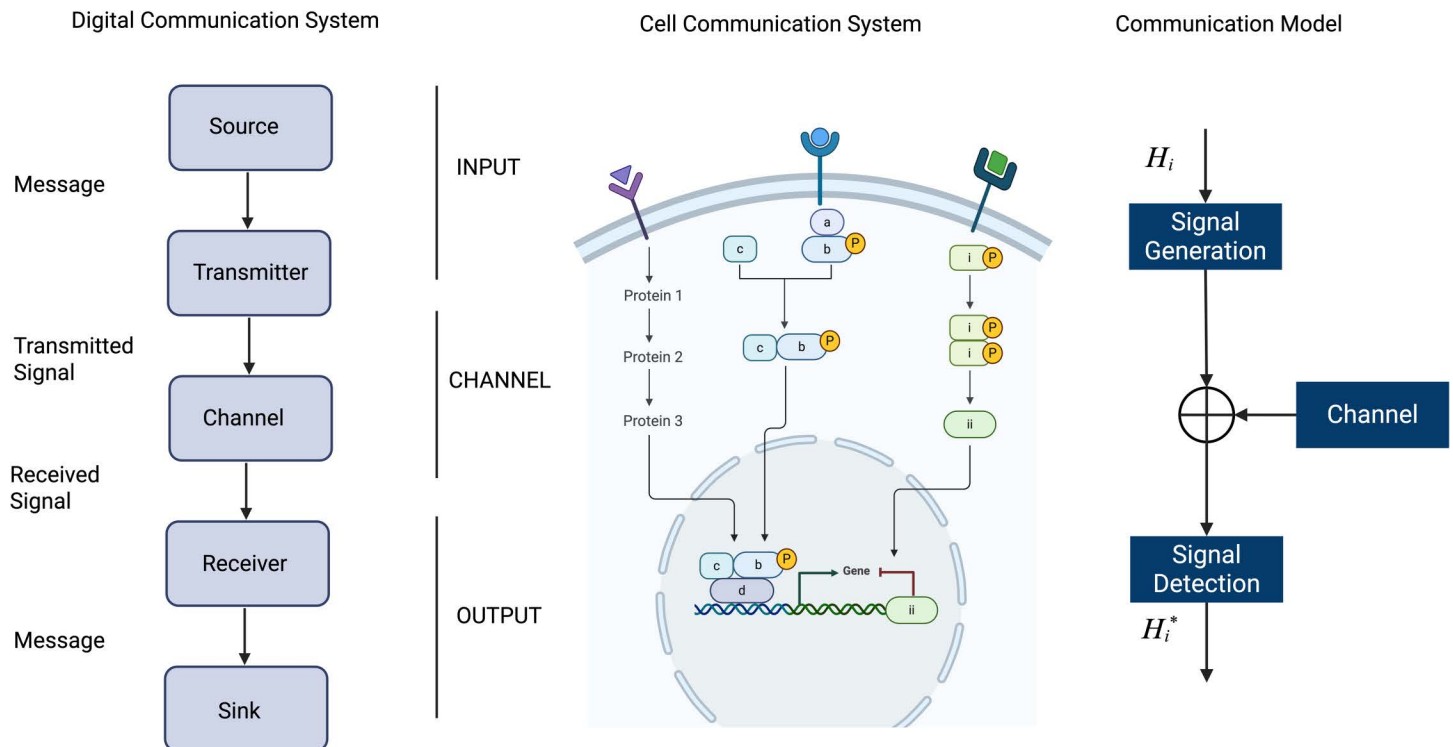

**Fig 1. Elements of a communication system and signal detection model.** A digital communication system consisting of source, transmitter, channel, receiver, and sink is used to study a cellular communication system consisting of surface receptors, signaling pathways, transcription factors, and target gene expression. The communication system can be modeled as a signal detection problem consisting of signal generation and transmission through a channel before detection. Figure created with Biorender.

"correct" signal is the mean of the pSTAT responses, which is unique to each sample and stimulation. We then compare characteristics of signal detection and identification in breast cancer samples to healthy donor controls to provide insights into immune dysfunction observed in breast cancer.

Unlike previous studies that applied information-theoretic concepts such as channel capacity or mutual information to determine how much information is transmitted through signaling pathways, we adopt the perspective of detection theory to determine the specificity of signaling networks. As such, we view cytokine discrimination as the following inference problem to be solved by the cell: *given the number of intracellular readout molecules, determine if an extracellular ligand is present, and decide on its type.* We note that this is not a determination of a "biologically correct" response, rather, a correct determination of the noisy signal as presented to the cell, which in an in vivo context would likely depend on cell extrinsic factors.

## Results

### Profiling signaling response

To characterize the specificity of immune signaling responses in estrogen receptor positive (ER+) breast cancer patients, we analyzed peripheral blood mononuclear cells (PBMCs) from 51 subjects, 32 of which were healthy donors (HD), and 19 of which were newly diagnosed with ER+ breast cancer (BC) (Tables 1, A and B in S1 Text). We systematically treated the PBMCs with 5 different cytokines alone or in combination with a JAK1/2 inhibitor ruxolitinib to study signaling responses. The cytokines included IL-2, IL-4, IL-6, IL-10, and IFNγ. After stimulation for 15 minutes, cells were fixed

**Table 1. Summary of data. Flow cytometry gating strategy provided in S1 Text.**

| Category | Details |
|---|---|
| Subjects | 51 total: 32 Healthy Donors (HD), 19 ER+ Breast Cancer (BC) patients |
| Sample Type | Peripheral Blood Mononuclear Cells (PBMCs) |
| Treatments | 5 cytokines (IL-2, IL-4, IL-6, IL-10, IFNγ) ± JAK1/2 inhibitor (Ruxolitinib) |
| Stimulation Time | 15 minutes |
| Analysis Method | Spectral Flow Cytometry |
| Markers Used | 27 intra- and extracellular markers |
| Signaling Transcription Factors | pSTAT1, pSTAT3, pSTAT4, pSTAT5, pSTAT6, pSMAD2/3 |
| Cell Populations | CD4 + T cells (TCM, TEM, naïve), CD8 + T cells (TCM, TEM, naïve), CD20 + memory B cells, naïve B cells (CD19 + CD27-), classical monocytes (CD14 + CD16-), NK cells (CD3-CD16+) |
| Minimum Cell Count for Inclusion | ≥ 200 cells per population |

and stained with a panel of 27 different intra- and extracellular markers, analyzed with spectral flow cytometry, and gated into distinct cell populations. Transcription factor phosphorylation was quantified by pSTAT1, pSTAT3, pSTAT4, pSTAT5, pSTAT6, and pSMAD2/3. Cell surface markers were used to identify CD4 + T cell subsets including central memory (TCM) CD4 + CD45RA-CD27 +, effector memory (TEM) CD4 + CD45RA-CD27-, and naïve T-cells CD4 + CD45RA+CD27+ (similarly for CD8 + T cells), CD20 + B cells, naïve B cells CD19 + CD27-, classical monocytes CD14 + CD16-, and CD3-CD16 + NK cells. Only cell populations with at least 200 cells were included in the analysis. The gating strategy to identify cell populations is provided in Figs A and B in S1 Text.

## Modeling ligand encoding and discrimination

Given the probabilistic nature of signal detection, we formulated the input signal (ligand) identification problem as a multi-hypothesis testing problem, where the receiver decides which of $M = 6$ signals is the input by choosing one of the following hypotheses $H_i$, $H_0$: baseline (no cytokine stimulation), $H_1$: stimulation with IL-4, $H_2$: stimulation with IL-2, $H_3$: stimulation with IL-10, $H_4$: stimulation with IL-6, or $H_5$: stimulation with IFNγ. The signal response space is the set of all responses (pSTAT1, pSTAT3, pSTAT4, pSTAT5, pSTAT6, pSMAD2/3) to all signals (Fig 2).

In order to compute the probability of correct ligand identification, we compute the probability of choosing the hypothesis $H_i$ given the true hypothesis $H_j$, $(Q_{ij})$. Then, the overall probability of error, $P_e$, is given by $P_e = \sum_{j=0}^{M-1} P(H_j) \sum_{i=0, \ i \neq j}^{M-1} Q_{ij}$ where P($H_j$) is the a priori probability of the hypothesis $H_j$. The optimal signal detector that minimizes the probability of error is called the maximum likelihood (ML) detector [17]. Details are provided in the Materials and Methods.

In addition to the overall probability of error, we used the signal-to-noise ratio (SNR) to characterize the specificity of the signaling response. In digital communication, SNR is a measure of fidelity of signal transmission and detection by receivers. There is a direct relationship between SNR and the probability of error in signal identification. Assuming a communication system with $M$ possible discrete signals transmitted under an additive white Gaussian noise channel, the measured response to a discrete signal $m$ is given by: $x = m + z$ where $z$ is a zero-mean Gaussian random variable with variance $\sigma^2$. The SNR is then given by $SNR = \frac{E[m^2]}{\sigma^2}$, where $E[m^2] = \frac{1}{M} \sum_{i=0}^{M-1} m_i^2$ if the signals are equiprobable.

## Examination of ligand identification error rates in healthy donors and ER+ breast cancer patients

We applied our communication model to calculate the probability of error and the signal-to-noise ratio, to characterize the specificity of ligand identification for each sample and cell type and compared BC to HD (Fig 3). This analysis reveals how

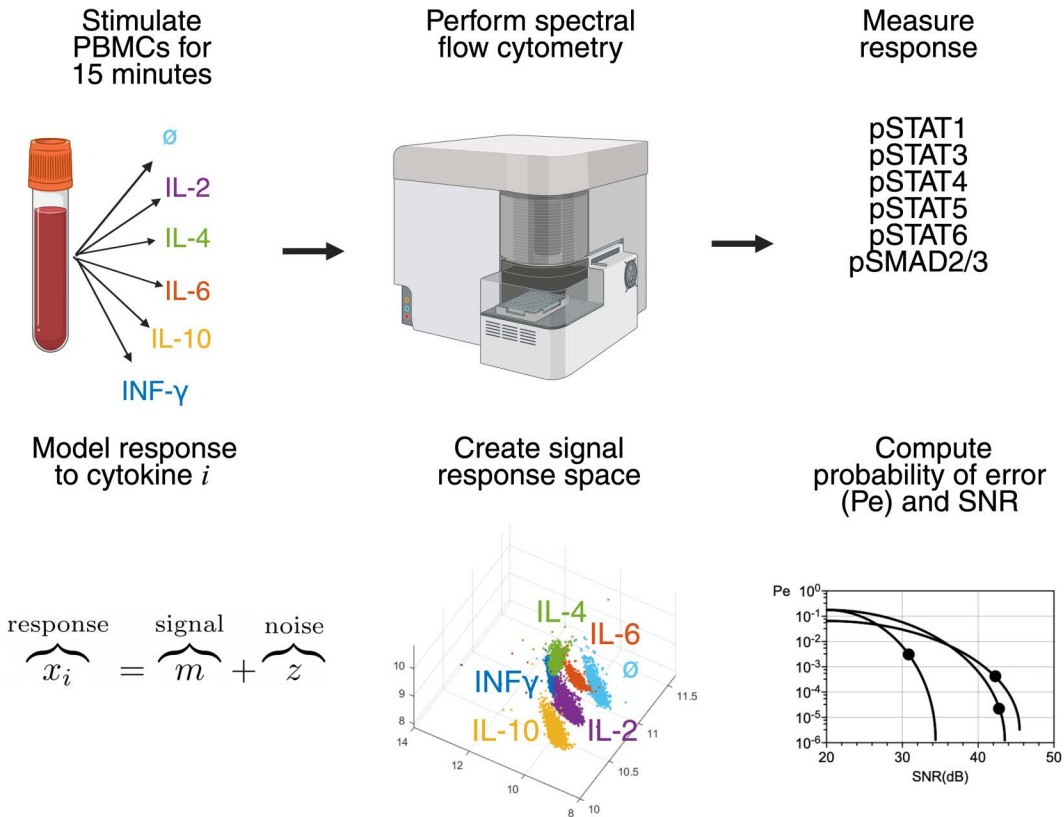

**Fig 2. Illustration of experiment, measurement, modeling, and analysis process.** Peripheral blood mononuclear cells from healthy donors and ER+ breast cancer patients were subjected to stimulation with one of 5 cytokines for 15 minutes. Cell surface markers and intracellular proteins were analyzed with spectral flow cytometry to establish cell identity and to measure response to cytokine stimulation. The response to any one of the cytokines is modeled as a sum of signal and noise. All stimulations and responses are combined to create a 6-dimensional signal response space, illustrated here in 3 dimensions. The probability of error, or signal misidentification ($P_e$), and signal-to-noise ratio (SNR) are computed from the signal response space for each sample and compared across cell types, healthy donors, and breast cancer patient samples. Figure created with Biorender.

immune cell populations differ in both the signal fidelity (SNR) and probability of signal identification error between BC and HD. Naïve CD4 + T cells and classical monocytes showed the largest error rates, with classical monocytes having lower SNR as compared to CD4 + T cells. Naïve CD8 + T cells showed lower SNR and higher error rates in BC as compared to HD. In contrast, classical monocytes show the same range of SNR for BC and HD but differ by an order of magnitude in the probability of error. Other cell types, such as naïve B cells, show overlapping SNR and error rates, with some healthy donor naïve B cells showing error rates exceeding those of breast cancer patients. NK cells showed the widest range of both SNR and error rates for both HD and BC. We observed increased error rates and reduced SNR for BC as compared to HD for nearly all immune cell subtypes (Fig 3B). SNR and $P_e$ graphs for CD4 +, CD8 + T, and B cell subsets are provided in Figs C-E in S1 Text.

## Immune signaling error profiles

In order to compare the profile of signaling error rates across all cell types in ER+ breast cancer patients, we plotted the SNR and $P_e$ values together on common SNR-$P_e$ axes (Fig 4A). When compared directly, the cells grouped into 4 distinct regions, consisting of combinations of low and high SNR and $P_e$, with classical monocytes showing low signal and high error, CD4 + T cells and central and effector memory subsets showing high signal and high error, B cells (naïve and

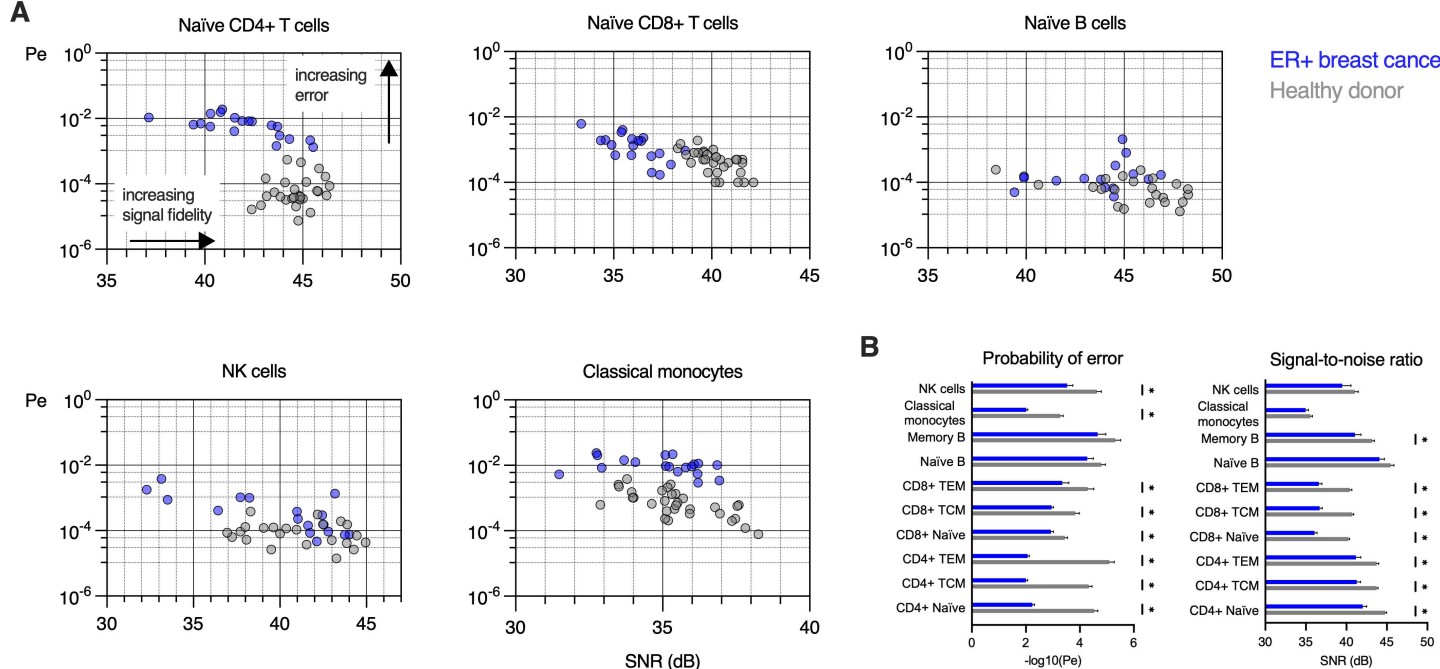

**Fig 3. Probability of error and signal-to-noise ratio are altered in peripheral blood immune cells in ER+ breast cancer patients as compared to healthy donors. A.** Signal detection is characterized by $P_e$ and SNR for naïve CD4+, CD8+, T cells, naïve B cells, NK cells, and classical monocytes in peripheral blood samples from 19 ER+ breast cancer patients and 32 healthy donors. Each datapoint corresponds to one healthy donor (grey) or ER+ breast cancer (blue) sample, integrating 6 different phosphorylation events from each of the 5 cytokine stimulations and baseline (no stimulation). **B.** Pairwise comparisons of the negative log transformed probability of error ($-\log_{10}(P_e)$) and SNR for HD and BC for each cell subtype (unpaired t-tests adjusted for multiple comparisons, ∗$p < 0.05$). For illustration in bar graphs, samples with $P_e = 0$ have $-\log_{10}(P_e) := 6$.

memory) showing high signal and low error, and CD8+T cells (including TCM and TEM) showing low signal and low error. To quantitatively compare these profiles for both BC and HD samples, we performed hierarchical clustering on the negative log transformed $P_e$ values (Fig 4B). This analysis reveals a clear separation between BC and HD in terms of signaling specificity (columns), with a higher overall probability of signal misidentification in BC compared to HD, as well as similarity of cell types (rows), with B, CD4+, and CD8+ cells clustering together. This analysis reveals patterns of signal detection errors across immune cell types that are similar across HD and BC samples.

In addition to the probability of error, we examined error rates for pairs of ligands, which gives rise to a confusion matrix. For many cell types, the confusion analysis resulted in small variations for most pairs of ligands, however, the analysis revealed that IL-2 and IL-4 pair-wise error rates dominate the overall probability of error in naïve CD4+T cells. In other words, IL-2 and IL-4 are more likely to be confused one for another compared to the other cytokine pairs. The analysis also indicates a trend in elevated pair-wise error rates between IL-2 and IL-4 in CD4+TCM cells and NK cells (S1 Figs F-K in S1 Text). No statistically significant correlations were found between signaling error rates or SNR with patient age, tumor stage, or receptor expression.

## Signaling detection alterations induced by JAK inhibition

In order to investigate the impact of signaling kinase inhibition on ligand identification error rates and signal fidelity, we treated 10 of the healthy donor samples with a clinically approved JAK1/2 inhibitor, ruxolitinib. JAK1/2 inhibition increased the identification error rate and decreased the signal fidelity (increased noise) in all immune cell subtypes, resulting in a

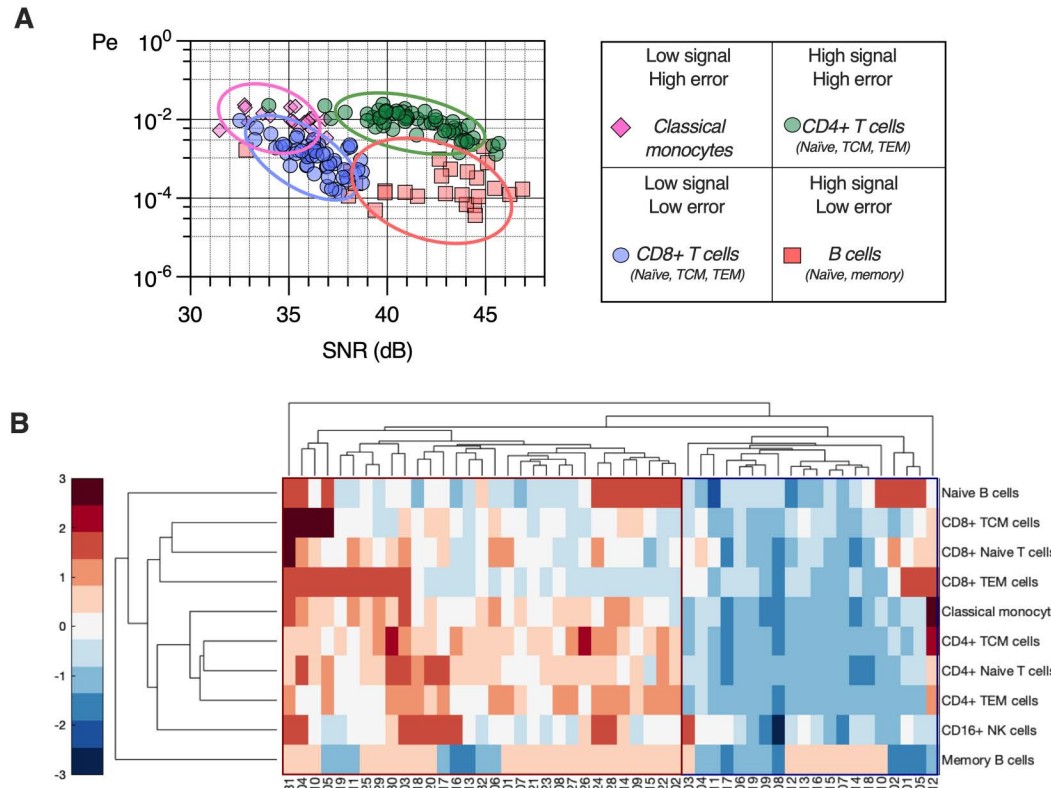

**Fig 4. Patterns of signaling error rates and SNR across immune cells in ER+ breast cancer. A.** Immune cells cluster in $P_e$ and SNR by type and can be grouped into low and high signal and error combinations. **B.** Hierarchical clustering of signaling error across immune cell subtypes reveals patterns of error rates (columns) and increased error rates across immune cell subtypes (rows) that distinguish breast cancer samples from healthy donors. Colorbar indicates row-wise z-score.

pattern of signaling dysregulation distinct from both controls and BC samples (Fig 5A). Ruxolitinib increased the cell-type specific error rate in several cell types beyond those observed in BC samples, including classical monocytes, memory and naïve B cells, and CD8+T cells, with the notable exception of Naïve CD4+T cells, which showed a drastic decrease in signal fidelity but no change in error rate. JAK1/2 inhibition resulted in decreased SNR for all cell types as compared to BC and control samples (Fig 5C,D).

To compare the immune cell types to each other, we plotted the ruxolitinib samples together on common SNR-$P_e$ axes (Fig 5B). This analysis revealed the stark differences in effect of the ruxolitinib by cell type, with classical monocytes and CD8+T cells (including naïve, TEM, and TCM subsets) having the largest error rates and smallest SNR. In contrast, CD4+T cells (including naïve, TEM, and TCM subsets) showed the smallest error rate and maintained a relatively large SNR.

## Discussion

We have presented a communication model to quantify signal fidelity and error rates based on digital communication theory and shown that immune cells in the peripheral blood of ER+ breast cancer patients exhibit increased ligand identification error rates and reduced signal fidelity as compared to healthy controls. We observed patterns of ligand detection error rates across immune cells, suggesting that a higher probability of ligand misidentification in BC patients may be an

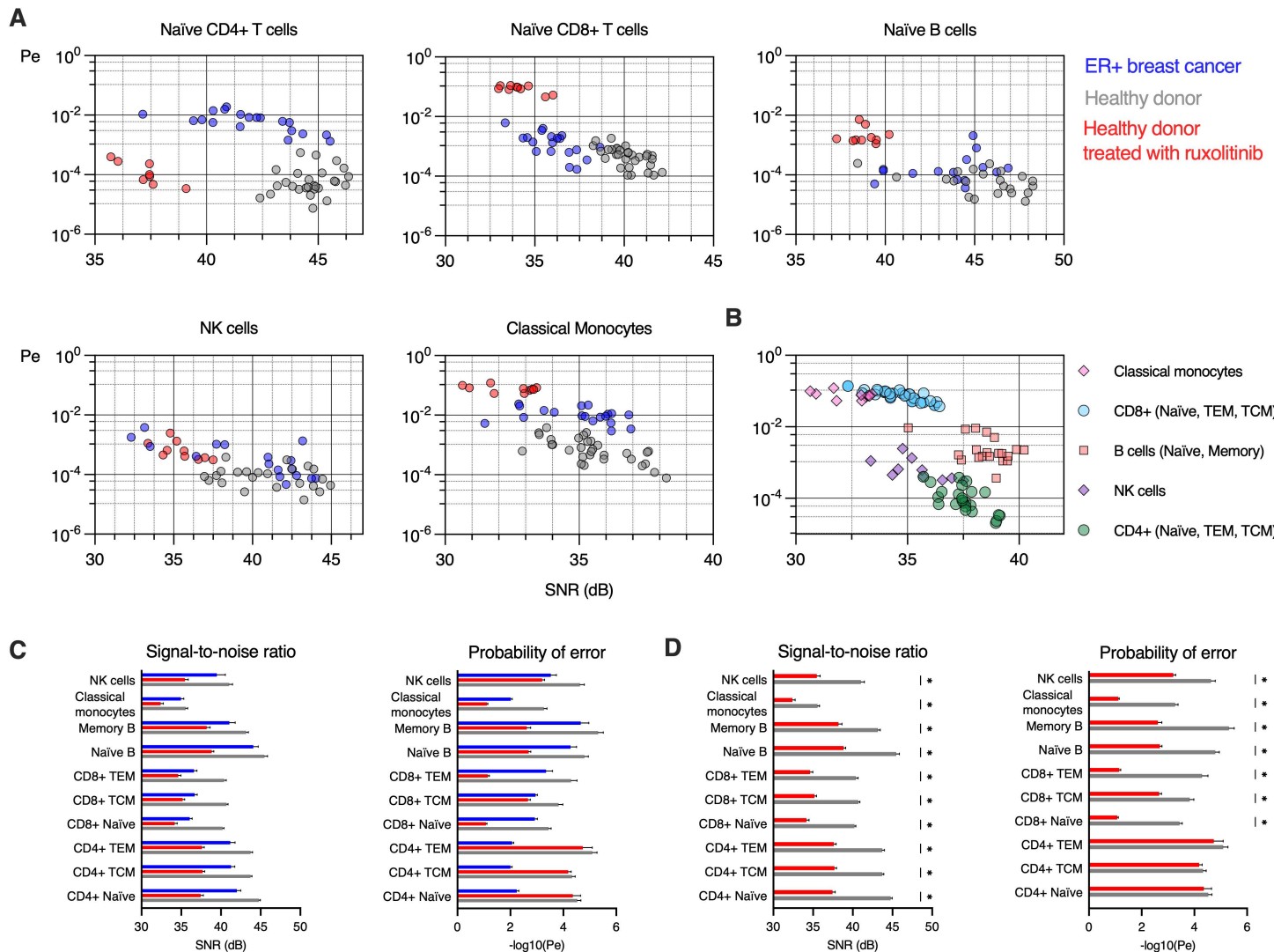

**Fig 5. JAK1/2 inhibition induces error rates and reduces SNR in healthy donors to levels comparable to breast cancer. A.** Immune cell subtypes for ER+ breast cancer samples (blue), healthy donors (grey) and a subset of 10 healthy donor samples also treated with ruxolitinib (red). Signal detection error rates are increased, and signal fidelity is decreased in all cell subtypes except for the error rate in naïve CD4 + T cells. **B.** Plotting the 10 samples treated with ruxolitinib on the same SNR-$P_e$ axes reveal relative error and signal fidelity characteristics. **C.** Comparisons of SNR and error rates for all cell subtypes in BC, HD, and HD + rux, and **D.** comparison of HD and HD + rux (unpaired t-tests adjusted for multiple comparisons, *p < 0.05). For illustration in the bar graphs, samples with $P_e = 0$ have $-\log_{10}(P_e) := 6$.

indication of altered immune signaling, potentially leading to immune dysfunction in BC patients. This leads us to hypothesize that BC patients that have lower signal detection error rates could have better prognoses through improved immune system functioning. In contrast, signaling kinase inhibiting therapies such as clinically approved JAK inhibitors may induce immune system dysregulation through reduction of SNR and amplification of $P_e$, which could be unintended downstream consequences of JAK suppression [7]. Specifically, we observed that JAK1/2 inhibition resulted in a pattern of immune signaling dysregulation different from controls or BC samples, with statistically significant increases in error rates and decreases in SNR observed in all cell types as compared to BC samples, with the notable exception of the error rates in CD4 + T cell subsets as relative to controls (Figs 5D and P in S1 Text).

Although prior studies have applied concepts from information and communication theory to characterize cell signaling [9,12,13,18–23], this work is the first to provide a measurement of ligand detection probability as a function of the signal-to-noise ratio by integrating the response of multiple transcription factors to multiple stimuli, thus enabling a more holistic view of signaling responses across multiple cell types.

Several prior experimental and theoretical studies have shown that pSTAT abundance and signal transduction peak around 15 minutes following cytokine stimulation [24–28]. We therefore hypothesized that signal detection accuracy should be maximized, and consequently, the probability of error minimized, at or around this timepoint. We note that signaling errors may be detected and subsequently corrected after STAT phosphorylation events and prior to protein production or other functional events such as proliferation, exhaustion, or microenvironmental factors. As a potential explanation for the signaling errors we observed, we explored the potential connection between receptor expression and SNR and $P_e$ through correlation analysis, as several prior studies have identified receptor dimerization to be a key mechanism of ligand discrimination [8,29–34]. We found no consistent patterns across cell types or receptors for either healthy donors or BC samples that would explain differences observed in either SNR or $P_e$ (Figs L-O in S1 Text).

A limitation of this work is the focus on intracellular signaling events, without analysis of their downstream consequences, such as proliferation or T cell exhaustion. As a foundational study, our primary aim was to establish the validity and applicability of this communication modeling framework to the well-characterized first steps of the cell signaling process in a controlled experimental setting. Because the detection error rate is calculated by integrating the signaling response of 5 cytokine stimulations individually, we interpret the error rate as the probability that a cell incorrectly identifies an isolated signal/ligand and does not consider the case when multiple ligands are presented simultaneously, as is likely the case *in vivo*. Moreover, the study of cell response to cytokine stimulation in an ex vivo context may not reflect response in an in vivo context. It is difficult, perhaps technically impossible at the time of this study, to measure the response of a cell to multiple simultaneous signals in an *in vivo* setting. However, our observations of orders of magnitude differences in SNR and $P_e$ in several cell types suggest cell intrinsic mechanisms of signaling alterations in BC patients. As such, investigation into the functional outcomes of signaling errors, the establishment of a critical error rate, and the application of these methods to an *in vivo* setting remain outstanding questions and form the foundation for future research.

Although causality cannot be inferred from our data, we reason that signaling alterations observed in ER+BC patients compared to healthy controls may be explained by inflammatory and cellular stress mechanisms induced by the presence of breast cancer. Elevated blood levels of IL-6 and IL-10 in ER+BC suggest a pro-inflammatory environment that can chronically activate pathways such as IL-6/STAT3, which are known to correlate with poorer survival outcomes and may reflect more aggressive disease states [1,2,35]. Prolonged exposure to elevated IL-6 and other cytokines circulating in the peripheral blood can induce changes in cell membranes, potentially affecting signal transduction without altering receptor expression [36]. This immune stress and persistent inflammation may contribute to increased cellular entropy and systemic immune signaling dysregulation, which are known hallmarks of cancer [37]. Together, these factors may create a milieu in which signaling pathways are disrupted, even in the absence of changes in receptor levels, highlighting the impact of cytokine-driven stress and tumor-induced disruption and entropy in cellular communication, resulting in changes to SNR and increased signal detection error rates.

One significant finding of our analysis is the contrast in signaling characteristics between CD4+ and CD8+T cells. CD4+T cells, including naïve, effector, and central memory subsets, stand out as having the highest error rates in BC, despite also showing large SNR, and yet, CD4+T cells appear to be the least sensitive to JAK1/2 inhibition, with a reduced SNR but no change in the detection error rate. An intriguing aspect of this analysis is the pairwise error analysis which reveals the elevated potential for IL-2 versus IL-4 signal misidentification in CD4+T cells in breast cancer patients. In comparison, CD8+T cells have a low SNR and low error rate, and yet are extraordinarily sensitive to

JAK1/2 inhibition, with orders of magnitude increase in error rate and the most drastic decrease in SNR. We hypothesize that due to their cytolytic function, CD8+T cells require more stringent activation signals involving co-stimulation to avoid killing healthy cells, leading to more refined signal transduction pathways. This would correspond to the observed lower SNR and higher detection error rates for isolated, single cytokine stimulations in our experimental conditions. This is in contrast to CD4+T cells, which differentiate into multiple helper subsets and therefore may have more broad functional consequences of signaling, corresponding to more specific responses to single cytokine stimulations with higher SNR and lower detection error rates. These dynamics suggest complex signaling interplay between T cell subsets that may provide avenues for the identification of patients with favorable immune signaling profiles or for therapeutic targeting.

## Materials and methods

### Ethics statement

Peripheral blood samples were obtained from breast cancer patients treated at City of Hope National Medical Center in Duarte, California, in compliance with protocols approved by the Institutional Review Board with informed consent (IRB 21368 and 19186).

### Human peripheral blood samples

The study cohort consisted of individuals with newly diagnosed breast cancer, all of whom were estrogen receptor-positive (ER+), progesterone receptor-positive (PR+) and HER2/neu receptor-negative (HER2-), grade Ia-IIb, with a mean age of 57.3 years at diagnosis (range 35–76 years). All samples were obtained prior to treatment. TNM tumor staging and percent positive ki67 staining were collected for each patient. A summary of patient characteristics is provided in Table A in S1 Text. Blood was collected in EDTA-treated tubes. Peripheral blood mononuclear cells were subsequently isolated using Ficoll-Paque density gradient centrifugation (Cytiva, Marlborough, MA, USA), following the manufacturer's protocol. The isolated PBMCs were cryopreserved in a solution containing 10% dimethyl sulfoxide (DMSO) and fetal bovine serum (FBS). Age-matched healthy control samples were acquired from the City of Hope Blood Donor Center.

### Cell culture

Cryopreserved PBMCs were carefully thawed and incubated overnight (16 hours) in RPMI 1640 medium, enriched with 10% fetal bovine serum and 1% penicillin-streptomycin-glutamine (PSG), under controlled conditions (37°C, 5% CO2). Cell counts and viability assessments were conducted using a hemocytometer and trypan blue exclusion method (Sigma-Aldrich). Subsequently, the cells were cultured in a 96 deep-well plate at densities ranging from 0.5 to $1 \times 10^6$ cells/ml in fresh RPMI 1640 medium (Thermo Fisher Scientific Inc., MA, USA).

### Cytokine stimulation

Following a resting period, PBMCs were cultured either untreated, stimulated with cytokine alone, or in combination with 0.1 mmol/L ruxolitinib (Cayman Chemical, Ann Arbor, MI, USA), a selective Janus kinase (JAK) 1/2 inhibitor. Cytokines included IFNγ (50 ng/ml), IL-10 (50 ng/ml), IL-2 (50 ng/ml), IL-6 (50 ng/ml), or IL-4 (50 ng/ml) (PeproTech, Rocky Hill, NJ, USA) at 37°C for 15 minutes. Following stimulation, cells were fixed with 1.5% paraformaldehyde (PFA) for 10 minutes at room temperature to preserve cellular structures and signaling intermediates. Fixed cells were then washed with phosphate-buffered saline (PBS) and permeabilized using ice-cold 100% methanol. Methanol-treated cells were stored at -80°C until further analysis. Before antibody staining, the fixed and permeabilized cells were washed three times with staining buffer (PBS supplemented with 1% fetal bovine serum).

## Phospho flow cytometry

Phospho flow cytometry was performed using the following antibodies: STAT4-AF647 (clone 38/p-Stat4), CD14-APC-Cy7 (clone HCD14), CD20-AF700 (clone H1), STAT6-V450 (clone 18/pStat6), PD-L1-BV510 (clone 29E.2A3), CD3-BV570 (clone UCHT1), PD1-BV605 (clone EH12.1), CD33-BV750 (clone p67.6), CD27-BV786 (clone L128), CD45RA-BUV395 (clone HI100), CD4-BUV563 (clone SK3), CD16-BUV737 (clone 3G8), CD8-BUV805 (clone SK1), STAT3-AF488 (clone 4/p-Stat3), STAT1-Percp-Cy5.5 (clone 4a), SMAD2/3-PE (clone O72-670), Foxp3-PE-CF594 (clone 259D/C7), and STAT5-PE-Cy7 (clone 47). Antibody dilutions were prepared according to the manufacturer's instructions and optimized through preliminary experiments to achieve optimal staining. The incubation was conducted for 45 minutes at room temperature. All antibodies were sourced from BioLegend (San Diego, CA, USA) or BD Biosciences (Franklin Lakes, NJ, USA).

## Data acquisition and gating strategy

Stained cells were analyzed using a Cytek Aurora flow cytometer, equipped with lasers at 355 nm, 405 nm, 488 nm, 561 nm, and 640 nm. Compensation settings were established using single-stain controls along with a negative control. Data acquisition was conducted at a rate of 1000 events per second, with between 50,000 and 100,000 events collected per sample. Gating strategies for cell population identification are provided in Figs A and B in S1 Text

## Data processing

Because we observed that pSTAT or pSMAD measurements did not always follow a Gaussian distribution, with their observed distribution tending closer to a log-normal distribution, all pSTAT/pSMAD measurements were first transformed with a log-like transformation (eq. 1). Then the transformed measurements were fit to a Gaussian mixture (GM) with two components. Each component is a multivariate normal distribution, where each variate corresponds to one of the 6 signaling molecules (5 pSTATs and pSMAD2/3). The log-transformed measurements together with knowledge of the GM parameters were fed to the optimal detector in a Monte-Carlo style simulation to obtain the probability of correct signal detection, or equivalently, the average probability of error in each analyzed signaling system. Here, one signaling system refers to the signaling pathway corresponding to one HD or BC patient and one cell type.

All pSTAT/pSMAD response data were log transformed as:

$$x_i(k) = \log\left(d_i(k) - \alpha_i(k) + 1\right), \ 0 \leq i \leq M-1, \quad 1 \leq k \leq K \tag{1}$$

where $d_i(k)$ is the measured response of the $k$th pSTAT/pSMAD to the $i$th treatment, $M = 6$ is the number of treatments (5 cytokines and untreated case), and $\alpha_i(k)$ is a parameter found by taking the minimum over all $d_i(k)$ measurements belonging to a given cell type and donor category (HD or BC). A global offset $\alpha_i(k) - 1$ was introduced to ensure the argument of the log function is greater or equal to one, since $d_i(k) - \alpha_i(k) \geq 0$.

Measurement data was organized in comma separated value (.csv) files, one file per subject, cytokine treatment, and cell type, provided in S1 Text. Given the large amount of data, the measurements were consolidated in structure arrays, one per HD/BC subject, cell type, and cytokine treatment. Each field in the structure array is a matrix with $K = 6$ columns (one column for each pSTAT/pSMAD) and $N_i$ rows, where $N_i$ is the number of gated cells for the $i$th cytokine treatment.

## Gaussian mixture modeling

GM modeling assumes that the log-transformed responses have a multi-variate Gaussian mixture distribution with two components. That is, the probability density function (PDF) of $\boldsymbol{x_i} = [x_i(1), \ x_i(2), \ldots x_i(K)]$ is given by:

$$p\left(\boldsymbol{x_i}\right) = \rho_i \aleph\left(\boldsymbol{m}_{1,i}, \boldsymbol{\Sigma}_{1,i}\right) + (1 - \rho_i) \, \aleph\left(\boldsymbol{m}_{2,i}, \boldsymbol{\Sigma}_{2,i}\right), \tag{2}$$

where $\aleph\left(\boldsymbol{m}_{1,i}, \Sigma_{1,i}\right)$ and $\aleph\left(\boldsymbol{m}_{2,i}, \Sigma_{2,i}\right)$ are multi-variate Gaussians with means $\boldsymbol{m}_{1,i}$ and $\boldsymbol{m}_{2,i}$, and covariance matrices $\Sigma_{1,i}$ and $\Sigma_{2,i}$ respectively, and $0 \leq \rho_i \leq 1$ is the mixing parameter. The means, covariance matrices, and mixing parameters were obtained from the log-transformed measurements using the MATLAB function "*fitgmdist*," which implements the Expectation-Maximization algorithm.

**Probability of error computation**

In order to compute the probability of error, let $Q_{ij} = P(H_i|H_j)$ be the probability of choosing the hypothesis $H_i$ given the true hypothesis $H_j$. Then, the overall probability of error, $P_e$, is given by $P_e = \sum_{j=0}^{M-1} P(H_j) \sum_{i=0, i \neq j}^{M-1} Q_{ij}$ where $P(H_j)$ is the a priori probability of the hypothesis $H_j$. The optimal signal detector that minimizes the probability of error under the Bayes strategy is one that maximizes the probability of observed data under the hypothesis $H_i$. More precisely, suppose that $\boldsymbol{x}$ is the multi-variate observation and $p(\boldsymbol{x}|H_i)$ are the conditional probability density functions (PDFs) of $\boldsymbol{x}$ under $H_i$. Moreover, assume equiprobable hypotheses, i.e., $P(H_i) = 1/M$, which is a reasonable assumption in the absence of prior information on the frequencies of $H_i$. Then, the optimal detector decides $H_i$ if $p(\boldsymbol{x}|H_i)$ is maximized. That is, it decides $H_i$ if $p(\boldsymbol{x}|H_i) > p(\boldsymbol{x}|H_j)$, for all $j \neq i$. This detector is called the maximum likelihood (ML) detector [17].

To evaluate the average error probability for one sample and one cell type, all the log-transformed pSTAT/pSMAD responses for all hypotheses ($M$ matrices $\boldsymbol{X}_i$ of size $N_i \times K$, $0 \leq i \leq M-1$) were fed into a simulator that implements the ML detector and accumulates the number of detection errors over all the simulated trials ($N = \sum_{i=0}^{M-1} N_i$). In this simulator implementation, one trial corresponds to a single cell response for each hypothesis, i.e., a row in the $\boldsymbol{X}_i$ matrices. The simulation loops over each hypothesis $H_j$, $0 \leq j \leq M-1$ and each trial $n$, $1 \leq n \leq N_j$, and computes the conditional probability of the hypothesis $H_j$ (the measured response $\boldsymbol{X}_j(n)$) given the hypothesis $H_i$, $0 \leq i \leq M-1$. With the GM assumption, this conditional probability is given by:

$$p\left(H_j|H_i\right) = \rho_i p\left(\boldsymbol{X}_j(n)|\boldsymbol{m}_{1,i}\right) + (1-\rho_i)\left(\boldsymbol{X}_j(n)|\boldsymbol{m}_{2,i}\right)$$

where

$$p\left(\boldsymbol{X}_j(n)|\boldsymbol{m}_{1,i}\right) = \frac{1}{\sqrt{(2\pi)^M \det(\Sigma_{1,i})}} \exp\left\{-\frac{1}{2}\left(\boldsymbol{X}_j(n) - \boldsymbol{m}_{1,i}\right) \Sigma_{1,i}^{-1} \left(\boldsymbol{X}_j(n) - \boldsymbol{m}_{1,i}\right)^T\right\}$$

$$p\left(\boldsymbol{X}_j(n)|\boldsymbol{m}_{2,i}\right) = \frac{1}{\sqrt{(2\pi)^M \det(\Sigma_{2,i})}} \exp\left\{-\frac{1}{2}\left(\boldsymbol{X}_j(n) - \boldsymbol{m}_{2,i}\right) \Sigma_{2,j}^{-1} \left(\boldsymbol{X}_j(n) - \boldsymbol{m}_{2,i}\right)^T\right\}$$

$\boldsymbol{m}_{1,i}$ and $\boldsymbol{m}_{2,i}$ are the signals corresponding to the hypothesis $H_i$ and $\Sigma_{1,i}$ and $\Sigma_{2,i}$ are the noise covariance matrices corresponding to the hypothesis $H_i$.

Then, the ML detector selects the hypothesis $H_{i*}$ such that the conditional probability $p(H_j|H_i)$ is maximized:

$$i^* = argmax \left\{p\left(\boldsymbol{X}_j(n)|H_i\right), 0 \leq i \leq M-1\right\}.$$

If $i^* \neq i$, we declare a detection error and a counter in a $M \times M$ error matrix $\boldsymbol{Q}$ is incremented at the index $(i^*, i)$. Otherwise, a correct decision is made and the counter at the index $(i, i)$ is incremented in the error matrix. Finally, the conditional probability $Q_{ij}$, the probability of choosing the hypothesis $H_i$ given the true hypothesis $H_j$ is estimated from $Q_{ij} = \frac{\boldsymbol{Q}(i,j)}{N}$, and the overall probability of error $P_e$ is given by $P_e = 1 - P_c = 1 - \sum_{i=0}^{M-1} P(H_i) Q_{ii}$. Note that the confusion matrix is the matrix having the element indexed by $i$th row and $j$th column equal to the conditional probabilities $Q_{ij}$.

## Signal-to-noise ratio computation

SNR is a dimensionless ratio of signal power $P_S$ to noise power $P_N$. We extend the SNR definition to the GM mixture model as follows. We assume a communication system with $M$ possible signals which is described by the following relationship:

$$x = \begin{cases} m_1 + z_1 & \text{with probability } \rho \\ m_2 + z_2 & \text{with probability } 1 - \rho \end{cases}$$

That is, under the hypothesis $H_i$, the transmitted signal is equal to $m_{1,i}$ with probability $\rho_i$ and it is equal to $m_{2,i}$ with probability $1 - \rho_i$. Similarly, under the hypothesis $H_i$, the noise is $z_{1i}$ with probability $\rho_i$ and it is $z_{2i}$ with probability $1 - \rho_i$, where $z_{1,i}$ is a zero-mean Gaussian random variable with variance $\sigma_{1,i}^2$ and $z_{2i}$ is a zero-mean Gaussian random variable with variance $\sigma_{2,i}^2$.

Then, for a Gaussian mixture with two components, the SNR can be expressed as:

$$SNR = \frac{\sum_{i=0}^{M-1} \left( \rho_i m_{1,i}^2 + (1 - \rho_i) m_{2,i}^2 \right)}{\sum_{i=0}^{M-1} \left( \rho_i \sigma_{1,i}^2 + (1 - \rho_i) \sigma_{2,i}^2 \right)}.$$

Finally, we generalize the above expression for the multivariate signal case, where under the hypothesis $H_i$, the transmitted signal is equal to $m_{1,i}$ with probability $\rho_i$ and it is equal to $m_{2,i}$ with probability $1 - \rho_i$. The noise corrupting the signal $m_{1,i}$ is a zero-mean multivariate Gaussian noise with covariance matrix $\Sigma_{1,i}$, and the noise corrupting the signal $m_{2,i}$ is a zero-mean multivariate Gaussian noise with covariance matrix $\Sigma_{2,i}$. Under these assumptions, the SNR becomes:

$$SNR = \frac{\sum_{i=0}^{M-1} \left( \rho_i m_{1,i} m_{1,i}^T + (1 - \rho_i) m_{2,i} m_{2,i}^T \right)}{\sum_{i=0}^{M-1} \left( \rho_i Trace(\Sigma_{1,i}) + (1 - \rho_i) Trace(\Sigma_{2,i}) \right)}.$$

All communication model simulations and analyses were performed in MATLAB. Correlation and statistical analyses were performed in Prism Graphpad 10.4.0. Note that it is customary to express SNR in decibels (dB), where the relation between linear SNR and SNR in dB is given by $SNR_{dB} = 10 \log_{10} SNR$.

## Supporting information

**S1 Text. Table A.** Breast cancer patient characteristics. **Table B.** Healthy donor characteristics. **Fig A.** Identification of PBMCs, single cells, live cells, and T, myeloid, monocyte, and B cell populations. **Fig B.** Identification of T cell subsets. **Fig C.** Signal error and SNR for CD4 + T cell subsets. **Fig D.** Signal error and SNR for CD8 + T cell subsets. **Fig E.** Signal error and SNR for B cell subsets. **Fig F.** Confusion matrix for CD4 + TCM cells given by $-\ln(P_e)$. The most likely signal confusion is between IL-4 and IL-2 cytokine stimulation, followed by INF-γ and IL-2. **Fig G.** Confusion matrix for CD16 + NK cells given by $-\ln(P_e)$. The most likely signal confusion is between IL-4 and IL-2 cytokine stimulation. **Fig H.** Confusion matrix for CD8 + T cells given by $-\ln(P_e)$. The most likely signal confusion is between untreated and IL-10 IL-2 cytokine stimulation, followed by IL-2 and INF-γ. **Fig I.** Confusion matrix for naïve B cells given by $-\ln(P_e)$. The most likely signal confusion is between IL-6 and IL-10 cytokine stimulation, followed by IL-6 and IL-2. **Fig J.** Confusion matrix for classical monocytes given by $-\ln(P_e)$. The most likely signal confusion is between IL-6 and INF-γ cytokine stimulation, followed by IL-6 and IL-10. **Fig K.** Confusion matrix for CD20 + B cells given by $-\ln(P_e)$. The most likely signal confusion is between IL-10 and IL-2 cytokine stimulation. **Fig L.** Probability of error receptor expression correlation analysis. Receptor expression was correlated with Pe for each cell type in all HD

samples (n = 32, top) and BC samples (n = 19, bottom). Statistically significant positive correlations are shown in green and negative correlations are shown in red (∗$p < 0.05$, ∗∗$p < 0.01$, ∗∗∗$p < 0.001$, ∗∗∗∗$p < 0.0001$). **Fig M.** Signal-to-noise receptor expression correlation analysis. Receptor expression was correlated with SNR for each cell type in all HD samples (n = 32, top) and BC samples (n = 19, bottom). Statistically significant positive correlations are shown in green and negative correlations are shown in red (∗$p < 0.05$, ∗∗$p < 0.01$, ∗∗∗$p < 0.001$, ∗∗∗∗$p < 0.0001$). **Fig N.** Correlation analysis of receptor expression rSNR on classical monocytes with overall SNR in breast cancer samples. Statistically significant correlations were found in expression levels of CD130, CD119, CD122, CD210, and CD132 on classical monocytes in breast cancer samples (n = 19). CD130, CD119, and CD122 were positively correlated, and CD210 and CD132 were negatively correlated. Because overall SNR is computed by integrating all pSTAT responses from all cytokine stimulations, these correlations do not explain differences observed between healthy donors and BC samples; moreover, classical monocytes show the lowest SNR in HD and BC samples for any immune cell subtype. **Fig O.** Correlation analysis of PDL and PD1 expression rSNR on NK cells with Pe and overall SNR in healthy donors and breast cancer samples. Because of the clinical relevance of PDL1 and PD1 expression on immune cells, we examined the correlation of PD(L)1 expression levels in NK cells for healthy donors (n = 32) and breast cancer (n = 19) samples. Neither PDL1 nor PD1 were consistently correlated with Pe or SNR in either healthy donors or breast cancer samples. The significance of the correlation is driven by large outliers in Pe and small outliers in SNR. **Fig P.** Signaling profiles of ruxolitinib as compared to BC. Statistical comparisons of **A.** signal-to-noise ratio and **B.** error rates for all immune cell types for ruxolitinib (red) and breast cancer samples (blue) shown in main text Fig 5C (unpaired t-tests adjusted for multiple comparisons, ∗$p < 0.05$).
(DOCX)

## Acknowledgments

The authors acknowledge the participation of the blood donors and breast cancer patients that made this work possible and thank them for their contributions to the study.

## Author contributions

**Conceptualization:** Adina Matache, Konstancja Urbaniak, Sergio Branciamore, Andrei S. Rodin, Peter P. Lee, Russell C. Rockne.

**Data curation:** Adina Matache, Joao Rodrigues Lima-Junior, Peter P. Lee, Russell C. Rockne.

**Formal analysis:** Adina Matache, Maxim Kuznetsov, Konstancja Urbaniak, Sergio Branciamore, Andrei S. Rodin, Russell C. Rockne.

**Funding acquisition:** Andrei S. Rodin, Peter P. Lee, Russell C. Rockne.

**Investigation:** Adina Matache, Joao Rodrigues Lima-Junior, Maxim Kuznetsov, Sergio Branciamore, Peter P. Lee, Russell C. Rockne.

**Methodology:** Adina Matache, Joao Rodrigues Lima-Junior, Maxim Kuznetsov, Konstancja Urbaniak, Sergio Branciamore, Andrei S. Rodin, Russell C. Rockne.

**Project administration:** Andrei S. Rodin, Russell C. Rockne.

**Resources:** Joao Rodrigues Lima-Junior, Russell C. Rockne.

**Supervision:** Andrei S. Rodin, Peter P. Lee, Russell C. Rockne.

**Validation:** Adina Matache, Russell C. Rockne.

**Visualization:** Adina Matache, Russell C. Rockne.

**Writing – original draft:** Adina Matache, Russell C. Rockne.

**Writing – review & editing:** Adina Matache, Joao Rodrigues Lima-Junior, Maxim Kuznetsov, Konstancja Urbaniak, Sergio Branciamore, Andrei S. Rodin, Peter P. Lee, Russell C. Rockne.

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
