## [Decision Letter · Decision Letter 0]

22 May 2025

Ligand Discrimination in Immune Cells: Signal Processing Insights into Immune Dysfunction in ER+ Breast Cancer

PLOS Computational Biology

Dear Dr. Rockne,

Thank you for submitting your manuscript to PLOS Computational Biology. After careful consideration, we feel that it has merit but does not fully meet PLOS Computational Biology's publication criteria as it currently stands. Therefore, we invite you to submit a revised version of the manuscript that addresses the points raised during the review process.

Please submit your revised manuscript within 60 days Jul 22 2025 11:59PM. If you will need more time than this to complete your revisions, please reply to this message or contact the journal office at ploscompbiol@plos.org. Please include the following items when submitting your revised manuscript:

We look forward to receiving your revised manuscript.

Kind regards,

Marc Birtwistle

Section Editor

PLOS Computational Biology

**Journal Requirements:**

3) Thank you for including an Ethics Statement for your study. Please include:

i) The full name(s) of the Institutional Review Board(s) or Ethics Committee(s)

ii) Please state whether the consent obtained is verbal or written

5) We notice that your supplementary Figures, and Tables are included in the manuscript file. Please remove them and upload them with the file type 'Supporting Information'. Please ensure that each Supporting Information file has a legend listed in the manuscript after the references list.

Potential Copyright Issues:

i) Figures 1, and 2. We note that the figures are created through BioRender. Please confirm that you hold a Premium account and provide a pdf copy of the CC BY 4.0 Licence as provided by BioRender. For instructions on how to generate a CC BY 4.0 license for your figure, please see the guidelines here: https://help.biorender.com/hc/en-gb/articles/21282341238045-Publishing-in-open-access-resources. 

If you are using the free assets from BioRender, we are unable to publish these images as they are licenced under a stricter licence than CC BY 4.0. In this case we ask you to remove the BioRender images and replace them with open source alternatives.

See these open source resources you may use to replace images / clip-art:

- https://bioart.niaid.nih.gov/ 

- https://bioicons.com/

- https://healthicons.org/ 

- https://scidraw.io/

- https://reactome.org/icon-lib

- https://www.phylopic.org/images

7) We note that your Data Availability Statement is currently as follows: "All relevant data are within the manuscript and its Supporting Information files". Please confirm at this time whether or not your submission contains all raw data required to replicate the results of your study. Authors must share the “minimal data set” for their submission. PLOS defines the minimal data set to consist of the data required to replicate all study findings reported in the article, as well as related metadata and methods (https://journals.plos.org/plosone/s/data-availability#loc-minimal-data-set-definition).

8) Please amend your detailed Financial Disclosure statement. This is published with the article. It must therefore be completed in full sentences and contain the exact wording you wish to be published.

1) If the funders had no role in your study, please state: "The funders had no role in study design, data collection and analysis, decision to publish, or preparation of the manuscript."

9) Please provide a completed 'Competing Interests' statement, including any COIs declared by your co-authors. If you have no competing interests to declare, please state "The authors have declared that no competing interests exist". Otherwise please declare all competing interests beginning with the statement "I have read the journal's policy and the authors of this manuscript have the following competing interests:". 

**Reviewers' comments:**

Reviewer's Responses to Questions

Reviewer #1: This paper applies an information theoretic model to an experimental study of gene expression activation of immune cells under cytokine stimulation to understand immune dysfunction in ER+ breast cancer. The study is well motived given growing evidence for the central role of immune surveillance and its failure in cancer progression. The work provides a decent summary of relevant literature. There is prior work on information theoretic approaches to characterizing biological signaling networks of various kinds, but the approach here is still innovative and well differentiated from the prior work. The paper follows a reasonable strategy in modeling signaling in terms of a message (cytokine) that ultimately induce an output (pattern of gene expression activation) and seeking to understand how ability to correctly process that message is affected in cancers. The work focuses on a very specific model system, JAK/STAT signaling in ER+ breast cancers, and while it would be interesting to see the topic explored more broadly, the focus on a model system is understandably necessary given the experimental work involved. There are some important points of the approach that could be clearer, as explained below, but if we accept the model then the work leads to some interesting conclusions of immune dysfunction in cancers relative to healthy controls and of induction of cancer-like dysfunction through JAK/STAT inhibition. The results raise some questions of interpretation, however, also discussed in more detail below. Overall, it seems an interesting approach to an important topic although in need of at least more clarity and consideration of interpretation on some key points.

Major Criticisms:

1. My largest question flows from some confusion about exactly how the computational analysis is done. The model is explained at a high level and the use of information theoretic concepts of signal to noise ratio (SNR) and error rate to quantify immune function seem sensible. But I did not follow where the canonical “correct” responses of each cell type to each stimulus are defined and are how they are applied. The model seems to imply that there is a correct output one expects to see with each stimulation, encoded as a set of ideal gene expression levels, but I did not see where these ground truth correct answers come from. Could the authors clarify?

2. A more focused technical question concerns the interpretation of the output as right or wrong based on whether the maximum likelihood assignment is the correct one. If one assumes the entire system is stochastic, then that output might itself be interpreted as a noisy input to downstream effects on gene regulation. If so, rather than saying that a stimulus produces one maximum likelihood output it might be more correct to say that a stimulus produces a probability density over possible outputs or a mixture of possible outputs. Subsequent analyses of SNR, error rate, and other information theoretic properties might be computed from this alternate interpretation much as they would be from the maximum likelihood one. I am not suggesting the authors need to redo the analysis with this alternative interpretation, but would ask them to consider it and whether it might affect the conclusions.

3. My next concern relates to the previous ones but is also more of a question of interpretation: should we expect there to be a canonical correct response from each cell type under each kind of stimulation or would we expect that to be more condition specific? If I understand the paper correctly, it is defining responses in cancers as errors or noise because they are not the responses one sees in healthy controls. But perhaps they are the “correct” responses for a different context, e.g., someone experiencing high systemic inflammation. Are responses in cancers noisier than those in healthy controls or are they the correct responses for a different context? Information theory should have tools to distinguish between those possibilities, in asking whether there is greater entropy of responses in cancer versus healthy or comparable entropy around a different mean. Perhaps that is what is done here, but if so, I just was not clear on that.

4. To broaden the prior questions, I felt there is an unexplored issue of causality here that could use further consideration and perhaps some additional exploration. If we accept that cancer patients, and cancer patients with bad outcomes, have more immune dysfunction than healthy controls, what is the causality? Does immune dysfunction lead to more diagnosable cancer and worse outcomes due to failures of surveillance? Are aggressive cancers evolving to disrupt the immune system in ways that lead to worse outcomes? Are patients who experience bad outcomes developing comorbidities that lead to more immune dysfunction? Are patients who are fighting aggressive cancers exhausting their immune systems in ways that lead to dysfunction? All of the above? The authors touch on some of this in the Discussion but I would like to see a more thorough consideration of biological significance of the results and alternative hypotheses. I would not expect the authors to be able to rule out all of these ideas experimentally, but I would be interested in knowing more about what they think is happening and why and whether their data can support or rule out any particular interpretations. If not, it would be interesting to see a bit more discussion about hypothetical future work by which one might get at the causality more effectively.

5. I also question whether some bias might be introduced by isolating PBMCs and effectively removing them from the natural context before conducting stimulation. Is there reason to be confident the response of isolated cells in culture would match that of the cells in situ? Even if not, it is still interesting that isolated cells from cancer patients behave differently from isolated cells from healthy controls, but perhaps it opens up other hypotheses for why there is a difference.

Minor critiques

6. I was confused by the text in the Discussion “This leads us to hypothesize that BC patients … in ways beyond simple suppression.” Could the authors elaborate on what was meant here?

7. Several of the figures use a particular plot style to show variability in error rates and SNRs (Figs 3A, 4A, 5A, 5B, S2, S5). The information conveyed by these plots is important but I have trouble distinguishing the colors of the spots without zooming in on an electronic copy to very high resolution, given the light coloring and the dark grid lines. I wonder if the authors might think of some way to make these plots larger and/or easier to see.

8. In Figure 5C/D, I am not sure I quite buy that the ruxolitinib-treated healthy samples look like cancer samples. The three conditions each look quite distinct, which is not surprising but I feel is not the message the authors are conveying. I think it would be useful to be a bit more precise about exactly what is claimed in interpreting these figures and how it can be established with statistical significance.

9. In Figure 5, the caption is missing a description for Fig 5D.

Reviewer #2: In this study, Matache, et al use an information theoretical approach to explore signal processing of immune cells captured in blood samples of a cohort of ER+ breast cancer patients and compare this to healthy controls. Consideration of how extracellular signals are encoded into responses, and how cells can discriminate between these signals is a unique approach not typically used. The analyses are comprehensive, but some of the biological rationale and interpretation seem overstated. Major and minor comments are outlined below.

Major comments

Raw data: The study would benefit from including an overview of the data. For example: cell type population frequency; ligand-induced signal change as compared to untreated

Confusion analysis: Many of the values in the confusion matrix are quite similar. The authors should comment on how biologically meaningful small differences are, and whether the ranks are over-interpreted. Based on prior literature about expected downstream signaling and impact of these cytokines, are the most commonly confused ligands expected?

JAK/STAT inhibition studies: Authors use the jak/stat inhibitor ruxolitinib “to investigate the clinical relevance of ligand identification error rates and signal fidelity”. The rationale for these studies is unclear. No malignant cells are being treated in this experiment, so the link to clinical relevance seems tenuous.

The interpretation of decreased signal fidelity is also lacking. One explanation for decreased signal fidelity in the presence of the inhibitor is that the inhibitor induced a strong pathway inhibition that cannot be overcome by ligand stimulation. Better casting these findings in the context of prior literature of how microenvironmental signals can influence therapeutic response would be beneficial.

Link to code base seems to be missing

Supplementary table of gated data are missing.

Minor comments

Figure S8 – figure legend does not seem to reflect the data presented

Figures S12-15 are only mentioned in the discussion. These should be mentioned in the main text

**Have the authors made all data and (if applicable) computational code underlying the findings in their manuscript fully available?**

Reviewer #1: Yes

Reviewer #2: **No: ** I looked for link to code and cannot find one. I also do not see a supplementary table of data (ie, gated cell calls), which authors indicated should be available.

PLOS authors have the option to publish the peer review history of their article (what does this mean? ). If published, this will include your full peer review and any attached files.

**Do you want your identity to be public for this peer review?** For information about this choice, including consent withdrawal, please see our Privacy Policy .

Reviewer #1: No

Reviewer #2: No

**Figure resubmission:**
---

## [Decision Letter · Decision Letter 1]

14 Oct 2025

Dear Dr. Rockne,

We are pleased to inform you that your manuscript 'Ligand Discrimination in Immune Cells: Signal Processing Insights into Immune Dysfunction in ER+ Breast Cancer' has been provisionally accepted for publication in PLOS Computational Biology.

Best regards,

Marc R Birtwistle, PhD

Section Editor

PLOS Computational Biology

Marc Birtwistle

Section Editor

PLOS Computational Biology

Reviewer's Responses to Questions

**Comments to the Authors: 
Please note here if the review is uploaded as an attachment.**

Reviewer #1: My previous concerns have been adequately addressed. The methodology is much clearer to me now. I think the paper also now does a better job of considering alternative interpretations and scaling back claims that might have been overreaching before. I do not have any other concerns to raise.

Reviewer #2: The authors have addressed my concerns.

**Have the authors made all data and (if applicable) computational code underlying the findings in their manuscript fully available?**

Reviewer #1: Yes

Reviewer #2: None

PLOS authors have the option to publish the peer review history of their article (what does this mean? ). If published, this will include your full peer review and any attached files.

**Do you want your identity to be public for this peer review?** For information about this choice, including consent withdrawal, please see our Privacy Policy .

Reviewer #1: No

Reviewer #2: No

---

## [Editor Report · Acceptance letter]

PCOMPBIOL-D-25-00518R1

Ligand Discrimination in Immune Cells: Signal Processing Insights into Immune Dysfunction in ER+ Breast Cancer

Dear Dr Rockne,

I am pleased to inform you that your manuscript has been formally accepted for publication in PLOS Computational Biology. Your manuscript is now with our production department and you will be notified of the publication date in due course.

With kind regards,

Anita Estes
